# Large Scale Canonical Correlation Analysis with Iterative Least Squares

**Yichao Lu**
University of Pennsylvania
yichaolu@wharton.upenn.edu

**Dean P. Foster**
Yahoo Labs, NYC
dean@foster.net

## Abstract

Canonical Correlation Analysis (CCA) is a widely used statistical tool with both well established theory and favorable performance for a wide range of machine learning problems. However, computing CCA for huge datasets can be very slow since it involves implementing QR decomposition or singular value decomposition of huge matrices. In this paper we introduce L-CCA, a iterative algorithm which can compute CCA fast on huge sparse datasets. Theory on both the asymptotic convergence and finite time accuracy of L-CCA are established. The experiments also show that L-CCA outperform other fast CCA approximation schemes on two real datasets.

## 1 Introduction

Canonical Correlation Analysis (CCA) is a widely used spectrum method for finding correlation structures in multi-view datasets introduced by [15]. Recently, [3, 9, 17] proved that CCA is able to find the right latent structure under certain hidden state model. For modern machine learning problems, CCA has already been successfully used as a dimensionality reduction technique for the multi-view setting. For example, A CCA between the text description and image of the same object will find common structures between the two different views, which generates a natural vector representation of the object. In [9], CCA is performed on a large unlabeled dataset in order to generate low dimensional features to a regression problem where the size of labeled dataset is small. In [6, 7] a CCA between words and its context is implemented on several large corpora to generate low dimensional vector representations of words which captures useful semantic features.

When the data matrices are small, the classical algorithm for computing CCA involves first a QR decomposition of the data matrices which pre whitens the data and then a Singular Value Decomposition (SVD) of the whitened covariance matrix as introduced in [11]. This is exactly how Matlab computes CCA. But for huge datasets this procedure becomes extremely slow. For data matrices with huge sample size [2] proposed a fast CCA approach based on a fast inner product preserving random projection called Subsampled Randomized Hadamard Transform but it's still slow for datasets with a huge number of features. In this paper we introduce a fast algorithm for finding the top $k_{\mathrm{cca}}$ canonical variables from huge sparse data matrices (a single multiplication with these sparse matrices is very fast) $\mathbf{X} \in n \times p_1$ and $\mathbf{Y} \in n \times p_2$ the rows of which are i.i.d samples from a pair of random vectors. Here $n \gg p_1, p_2 \gg 1$ and $k_{\mathrm{cca}}$ is relatively small number like 50 since the primary goal of CCA is to generate low dimensional features. Under this set up, QR decomposition of a $n \times p$ matrix cost $O(np^2)$ which is extremely slow even if the matrix is sparse. On the other hand since the data matrices are sparse, $\mathbf{X}^\top \mathbf{X}$ and $\mathbf{Y}^\top \mathbf{Y}$ can be computed very fast. So another whitening strategy is to compute $(\mathbf{X}^\top \mathbf{X})^{-\frac{1}{2}}, (\mathbf{Y}^\top \mathbf{Y})^{-\frac{1}{2}}$. But when $p_1, p_2$ are large this takes $O(\max\{p_1^3, p_2^3\})$ which is both slow and numerically unstable.

The main contribution of this paper is a fast iterative algorithm  L-CCA consists of only QR decomposition of relatively small matrices and a couple of matrix multiplications which only involves huge sparse matrices or small dense matrices. This is achieved by reducing the computation of CCA to a sequence of fast Least Square iterations. It is proved that  L-CCA asymptotically converges to the exact CCA solution and error analysis for finite iterations is also provided.  As shown by the experiments,  L-CCA also has favorable performance on real datasets when compared with other CCA approximations given a fixed CPU time.

It's worth pointing out that approximating CCA is much more challenging than SVD(or PCA). As suggested by [12, 13], to approximate the top singular vectors of $\mathbf{X}$, it suffices to randomly sample a small subspace in the span of $\mathbf{X}$ and some power iteration with this small subspace will automatically converge to the directions with top singular values.  On the other hand CCA has to search through the whole $\mathbf{X}$ $\mathbf{Y}$ span in order to capture directions with large correlation.  For example, when the most correlated directions happen to live in the bottom singular vectors of the data matrices, the random sample scheme will miss them completely.  On the other hand, what L-CCA algorithm doing intuitively is running an exact search of correlation structures on the top singular vectors and an fast gradient based approximation on the remaining directions.

## 2  Background: Canonical Correlation Analysis

### 2.1  Definition

Canonical Correlation Analysis (CCA) can be defined in many different ways. Here we use the definition in [9, 17] since this version naturally connects CCA with the Singular Value Decomposition (SVD) of the whitened covariance matrix, which is the key to understanding our algorithm.

**Definition 1.** *Let* $\mathbf{X} \in n \times p_1$ *and* $\mathbf{Y} \in n \times p_2$ *where the rows are i.i.d samples from a pair of random vectors. Let* $\mathbf{\Phi_x} \in p_1 \times p_1, \mathbf{\Phi_y} \in p_2 \times p_2$ *and use* $\phi_{x,i}, \phi_{y,j}$ *to denote the columns of* $\mathbf{\Phi_x}, \mathbf{\Phi_y}$ *respectively.* $\mathbf{X}\phi_{x,i}, \mathbf{Y}\phi_{y,j}$ *are called canonical variables if*

$$\phi_{x,i}^\top \mathbf{X}^\top \mathbf{Y} \phi_{y,j} = \begin{cases} d_i & if \quad i = j \\ 0 & if \quad i \neq j \end{cases}$$

$$\phi_{x,i}^\top \mathbf{X}^\top \mathbf{X} \phi_{x,j} = \begin{cases} 1 & if \quad i = j \\ 0 & if \quad i \neq j \end{cases} \qquad \phi_{y,i}^\top \mathbf{Y}^\top \mathbf{Y} \phi_{y,j} = \begin{cases} 1 & if \quad i = j \\ 0 & if \quad i \neq j \end{cases}$$

$\mathbf{X}\phi_{x,i}, \mathbf{Y}\phi_{y,i}$ *is the* $i^{th}$ *pair of canonical variables and* $d_i$ *is the* $i^{th}$ *canonical correlation.*

### 2.2  CCA and SVD

First introduce some notation. Let
$$\mathbf{C}_{xx} = \mathbf{X}^\top \mathbf{X} \quad \mathbf{C}_{yy} = \mathbf{Y}^\top \mathbf{Y} \quad \mathbf{C}_{xy} = \mathbf{X}^\top \mathbf{Y}$$
For simplicity assume $\mathbf{C}_{xx}$ and $\mathbf{C}_{yy}$ are full rank and Let
$$\tilde{\mathbf{C}}_{xy} = \mathbf{C}_{xx}^{-\frac{1}{2}} \mathbf{C}_{xy} \mathbf{C}_{yy}^{-\frac{1}{2}}$$
The following lemma provides a way to compute the canonical variables by SVD.

**Lemma 1.** *Let* $\tilde{\mathbf{C}}_{xy} = \mathbf{U}\mathbf{D}\mathbf{V}^\top$ *be the SVD of* $\tilde{\mathbf{C}}_{xy}$ *where* $u_i, v_j$ *denote the left, right singular vectors and* $d_i$ *denotes the singular values. Then* $\mathbf{X}\mathbf{C}_{xx}^{-\frac{1}{2}} u_i, \mathbf{Y}\mathbf{C}_{yy}^{-\frac{1}{2}} v_j$ *are the canonical variables of the* $\mathbf{X}$, $\mathbf{Y}$ *space respectively.*

*Proof.* Plug $\mathbf{X}\mathbf{C}_{xx}^{-\frac{1}{2}} u_i, \mathbf{Y}\mathbf{C}_{yy}^{-\frac{1}{2}} v_j$ into the equations in Definition 1 directly proves lemma 1 □

As mentioned before, we are interested in computing the top $k_{\text{cca}}$ canonical variables where $k_{\text{cca}} \ll p_1, p_2$. Use $\mathbf{U}_1, \mathbf{V}_1$ to denote the first $k_{\text{cca}}$ columns of $\mathbf{U}, \mathbf{V}$ respectively and use $\mathbf{U}_2, \mathbf{V}_2$ for the remaining columns. By lemma 1, the top $k_{\text{cca}}$ canonical variables can be represented by $\mathbf{X}\mathbf{C}_{xx}^{-\frac{1}{2}} \mathbf{U}_1$ and $\mathbf{Y}\mathbf{C}_{yy}^{-\frac{1}{2}} \mathbf{V}_1$.

---
**Algorithm 1** CCA via Iterative LS
---

    **Input :** Data matrix $\mathbf{X} \in n \times p_1$ , $\mathbf{Y} \in n \times p_2$. A target dimension $k_{\text{cca}}$. Number of orthogonal iterations $t_1$

    **Output :** $\mathbf{X}_{k_{\text{cca}}} \in n \times k_{\text{cca}}$, $\mathbf{Y}_{k_{\text{cca}}} \in n \times k_{\text{cca}}$ consist of top $k_{\text{cca}}$ canonical variables of $\mathbf{X}$ and $\mathbf{Y}$.

    1.Generate a $p_1 \times k_{\text{cca}}$ dimensional random matrix $\mathbf{G}$ with i.i.d standard normal entries.

    2.Let $\mathbf{X}_0 = \mathbf{XG}$

    3.

    **for** $t = 1$ **to** $t_1$ **do**

        $\mathbf{Y}_t = \mathbf{H_Y}\mathbf{X}_{t-1}$ where $\mathbf{H_Y} = \mathbf{Y}(\mathbf{Y}^\top\mathbf{Y})^{-1}\mathbf{Y}^\top$

        $\mathbf{X}_t = \mathbf{H_X}\mathbf{Y}_t$ where $\mathbf{H_X} = \mathbf{X}(\mathbf{X}^\top\mathbf{X})^{-1}\mathbf{X}^\top$

    **end for**

    4.$\mathbf{X}_{k_{\text{cca}}} = \text{QR}(\mathbf{X}_{t_1})$, $\mathbf{Y}_{k_{\text{cca}}} = \text{QR}(\mathbf{Y}_{t_1})$

    Function $\text{QR}(\mathbf{X}_t)$ extract an orthonormal basis of the column space of $\mathbf{X}_t$ with QR decomposition

---

## 3 Compute CCA by Iterative Least Squares

Since the top canonical variables are connected with the top singular vectors of $\tilde{\mathbf{C}}_{xy}$ which can be compute with orthogonal iteration [10] (it's called simultaneous iteration in [21]), we can also compute CCA iteratively. A detailed algorithm is presented in Algorithm1:

The convergence result of Algorithm 1 is stated in the following theorem:

**Theorem 1.** *Assume* $|d_1| > |d_2| > |d_3|... > |d_{k_{cca}+1}|$ *and* $\mathbf{U}_1^\top\mathbf{C}_{xx}^{\frac{1}{2}}\mathbf{G}$ *is non singular (this will hold with probability 1 if the elements of* $\mathbf{G}$ *are i.i.d Gaussian). The columns of* $\mathbf{X}_{k_{cca}}$ *and* $\mathbf{Y}_{k_{cca}}$ *will converge to the top* $k_{cca}$ *canonical variables of* $\mathbf{X}$ *and* $\mathbf{Y}$ *respectively if* $t_1 \to \infty$.

Theorem 1 is proved by showing it's essentially an orthogonal iteration [10, 21] for computing the top $k_{\text{cca}}$ eigenvectors of $\mathbf{A} = \tilde{\mathbf{C}}_{xy}\tilde{\mathbf{C}}_{xy}^\top$. A detailed proof is provided in the supplementary materials.

### 3.1 A Special Case

When $\mathbf{X}$ $\mathbf{Y}$ are sparse and $\mathbf{C}_{xx}, \mathbf{C}_{yy}$ are diagonal (like the Penn Tree Bank dataset in the experiments), Algorithm 1 can be implemented extremely fast since we only need to multiply with sparse matrices or inverting huge but diagonal matrices in every iteration. QR decomposition is performed not only in the end but after every iteration for numerical stability issues (here we only need to QR with matrices much smaller than $\mathbf{X}, \mathbf{Y}$). We call this fast version D-CCA in the following discussions.

When $\mathbf{C}_{xx}, \mathbf{C}_{yy}$ aren't diagonal, computing matrix inverse becomes very slow. But we can still run D-CCA by approximating $(\mathbf{X}^\top\mathbf{X})^{-1}, (\mathbf{Y}^\top\mathbf{Y})^{-1}$ with $(\text{diag}(\mathbf{X}^\top\mathbf{X}))^{-1}, (\text{diag}(\mathbf{Y}^\top\mathbf{Y}))^{-1}$ in algorithm 1 when speed is a concern. But this leads to poor performance when $\mathbf{C}_{xx}, \mathbf{C}_{yy}$ are far from diagonal as shown by the URL dataset in the experiments.

### 3.2 General Case

Algorithm 1 reduces the problem of CCA to a sequence of iterative least square problems. When $\mathbf{X}, \mathbf{Y}$ are huge, solving LS exactly is still slow since it consists inverting a huge matrix but fast LS methods are relatively well studied. There are many ways to approximate the LS solution by optimization based methods like Gradient Descent [1, 23], Stochastic Gradient Descent [16, 4] or by random projection and subsampling based methods like [8, 5]. A fast approximation to the top $k_{\text{cca}}$ canonical variables can be obtained by replacing the exact LS solution in every iteration of Algorithm 1 with a fast approximation. Here we choose LING [23] which works well for large sparse design matrices for solving the LS problem in every CCA iteration.

The connection between CCA and LS has been developed under different setups for different purposes. [20] shows that CCA in multi label classification setting can be formulated as an LS problem. [22] also formulates CCA as a recursive LS problem and builds an online version based on this observation. The benefit we take from this iterative LS formulation is that running a fast LS ap-

---
**Algorithm 2** LING
---
**Input :** $X \in n \times p$, $Y \in n \times 1$. $k_{\mathrm{pc}}$, number of top left singular vectors selected. $t_2$, number of iterations in Gradient Descent.

**Output :** $\hat{Y} \in n \times 1$, which is an approximation to $X(X^\top X)^{-1}X^\top Y$

1. Compute $U_1 \in n \times k_{\mathrm{pc}}$, top $k_{\mathrm{pc}}$ left singular vector of $X$ by randomized SVD (See supplementary materials for detailed description).

2. $Y_1 = U_1 U_1^\top X$.

3. Compute the residual. $Y_r = Y - Y_1$

4. Use gradient descent initial at the 0 vector (see supplementary materials for detailed description) to approximately solve the LS problem $\min_{\beta_r \in \mathcal{R}^p} \|X\beta_r - Y_r\|^2$. Use $\beta_{r,t_2}$ to denote the solution after $t_2$ gradient iterations.

5. $\hat{Y} = Y_1 + X\beta_{r,t_2}$.

---

proximation in every iteration will give us a fast CCA approximation with both provable theoretical guarantees and favorable experimental performance.

# 4  Algorithm

In this section we introduce L-CCA which is a fast CCA algorithm based on Algorithm 1.

## 4.1  LING: a Gradient Based Least Square Algorithm

First we need to introduce the fast LS algorithm LING as mentioned in section 3.2 which is used in every orthogonal iteration of L-CCA .
Consider the LS problem:
$$\beta^* = \arg\min_{\beta \in \mathbb{R}^p}\{\|X\beta - Y\|^2\}$$

for $X \in n \times p$ and $Y \in n \times 1$. For simplicity assume $X$ is full rank. $X\beta^* = X(X^\top X)^{-1}X^\top Y$ is the projection of $Y$ onto the column space of $X$. In this section we introduce a fast algorithm LING to approximately compute $X\beta^*$ without formulating $(X^\top X)^{-1}$ explicitly which is slow for large $p$. The intuition of LING is as follows. Let $U_1 \in n \times k_{\mathrm{pc}}$ ($k_{\mathrm{pc}} \ll p$) be the top $k_{\mathrm{pc}}$ left singular vectors of $X$ and $U_2 \in n \times (p - k_{\mathrm{pc}})$ be the remaining singular vectors. In LING we decompose $X\beta^*$ into two orthogonal components,
$$X\beta^* = U_1 U_1^\top Y + U_2 U_2^\top Y$$

the projection of $Y$ onto the span of $U_1$ and the projection onto the span of $U_2$. The first term can be computed fast given $U_1$ since $k_{\mathrm{pc}}$ is small. $U_1$ can also be computed fast approximately with the randomized SVD algorithm introduced in [12] which only requires a few fast matrix multiplication and a QR decomposition of $n \times k_{\mathrm{pc}}$ matrix. The details for finding $U_1$ are illustrated in the supplementary materials. Let $Y_r = Y - U_1 U_1^\top Y$ be the residual of $Y$ after projecting onto $U_1$. For the second term, we compute it by solving the optimization problem
$$\min_{\beta_r \in \mathbb{R}^p}\{\|X\beta_r - Y_r\|^2\}$$

with Gradient Descent (GD) which is also described in detail in the supplementary materials. A detailed description of LING are presented in Algorithm 2.
In the above discussion $Y$ is a column vector. It is straightforward to generalize LING to fit into Algorithm 1 where $Y$ have multiple columns by applying Algorithm 2 to every column of $Y$.
In the following discussions, we use LING $(Y, X, k_{\mathrm{pc}}, t_2)$ to denote the LING output with corresponding inputs which is an approximation to $X(X^\top X)^{-1}X^\top Y$.

The following theorem gives error bound of LING .

**Theorem 2.** *Use $\lambda_i$ to denote the $i^{th}$ singular value of $X$. Consider the LS problem*
$$\min_{\beta \in \mathbb{R}^p}\{\|X\beta - Y\|^2\}$$

---
**Algorithm 3** L-CCA
---

    **Input :** $\mathbf{X} \in n \times p_1$ ,$\mathbf{Y} \in n \times p_2$: Data matrices.

    $k_{\text{cca}}$: Number of top canonical variables we want to extract.

    $t_1$: Number of orthogonal iterations.

    $k_{\text{pc}}$: Number of top singular vectors for LING

    $t_2$: Number of GD iterations for LING

    **Output :** $\mathbf{X}_{k_{\text{cca}}} \in n \times k_{\text{cca}}$, $\mathbf{Y}_{k_{\text{cca}}} \in n \times k_{\text{cca}}$: Top $k_{\text{cca}}$ canonical variables of $\mathbf{X}$ and $\mathbf{Y}$.

    1.Generate a $p_1 \times k_{\text{cca}}$ dimensional random matrix $\mathbf{G}$ with i.i.d standard normal entries.

    2.Let $\mathbf{X}_0 = \mathbf{XG}$, $\hat{\mathbf{X}}_0 = \text{QR}(\mathbf{X}_0)$

    3.

    **for** $t = 1$ **to** $t_1$ **do**

        $\mathbf{Y}_t = \mathbf{LING}(\hat{\mathbf{X}}_{t-1}, \mathbf{Y}, k_{\text{pc}}, t_2)$, $\hat{\mathbf{Y}}_t = \text{QR}(\mathbf{Y}_t)$

        $\mathbf{X}_t = \mathbf{LING}(\hat{\mathbf{Y}}_t, \mathbf{X}, k_{\text{pc}}, t_2)$, $\hat{\mathbf{X}}_t = \text{QR}(\mathbf{X}_t)$

    **end for**

    4.$\mathbf{X}_{k_{\text{cca}}} = \hat{\mathbf{X}}_{t_1}$, $\mathbf{Y}_{k_{\text{cca}}} = \hat{\mathbf{Y}}_{t_1}$

---

for $X \in n \times p$ and $Y \in n \times 1$. Let $Y^* = X(X^\top X)^{-1}X^\top Y$ be the projection of $Y$ onto the column space of $X$ and $\hat{Y}_{t_2} = LING\,(Y, X, k_{pc}, t_2)$. Then

$$\|Y^* - \hat{Y}_{t_2}\|^2 \leq Cr^{2t_2} \tag{1}$$

for some constant $C > 0$ and $r = \frac{\lambda_{k_{pc}+1}^2 - \lambda_p^2}{\lambda_{k_{pc}+1}^2 + \lambda_p^2} < 1$

The proof is in the supplementary materials due to space limitation.

**Remark 1.** *Theorem 2 gives some intuition of why LING decompose the projection into two components. In an extreme case if we set $k_{pc} = 0$ (i.e. don't remove projection on the top principle components and directly apply GD to the LS problem), $r$ in equation 1 becomes $\frac{\lambda_1^2 - \lambda_p^2}{\lambda_1^2 + \lambda_p^2}$. Usually $\lambda_1$ is much larger than $\lambda_p$, so $r$ is very close to 1 which makes the error decays slowly. Removing projections on $k_{pc}$ top singular vector will accelerate error decay by making $r$ smaller. The benefit of this trick is easily seen in the experiment section.*

## 4.2 Fast Algorithm for CCA

Our fast CCA algorithm  L-CCA are summarized in Algorithm 3:

There are two main differences between Algorithm 1 and 3. We use LING to solve Least squares approximately for the sake of speed. We also apply QR decomposition on every LING output for numerical stability issues mentioned in [21].

## 4.3 Error Analysis of  L-CCA

This section provides mathematical results on how well the output of  L-CCA algorithm approximates the subspace spanned by the top $k_{\text{cca}}$ true canonical variables for finite $t_1$ and $t_2$. Note that the asymptotic convergence property of  L-CCA when $t_1, t_2 \to \infty$ has already been stated by **theorem 1**. First we need to define the distances between subspaces as introduced in section 2.6.3 of [10]:

**Definition 2.** *Assume the matrices are full rank. The distance between the column space of matrix $\mathbf{W}_1 \in n \times k$ and $\mathbf{Z}_1 \in n \times k$ is defined by*

$$dist(\mathbf{W}_1, \mathbf{Z}_1) = \|\mathbf{H}_{\mathbf{W}_1} - \mathbf{H}_{\mathbf{Z}_1}\|_2$$

*where $\mathbf{H}_{\mathbf{W}_1} = \mathbf{W}_1(\mathbf{W}_1^\top \mathbf{W}_1)^{-1}\mathbf{W}_1^\top$, $\mathbf{H}_{\mathbf{Z}_1} = \mathbf{Z}_1(\mathbf{Z}_1^\top \mathbf{Z}_1)^{-1}\mathbf{Z}_1^\top$ are projection matrices. Here the matrix norm is the spectrum norm. Easy to see $dist(\mathbf{W}_1, \mathbf{Z}_1) = dist(\mathbf{W}_1\mathbf{R}_1, \mathbf{Z}_1\mathbf{R}_2)$ for any invertible $k \times k$ matrix $\mathbf{R}_1, \mathbf{R}_2$.*

We continue to use the notation defined in section 2. Recall that $\mathbf{X}\mathbf{C}_{xx}^{-\frac{1}{2}}\mathbf{U}_1$ gives the top $k_{\text{cca}}$ canonical variables from $\mathbf{X}$. The following theorem bounds the distance between the truth $\mathbf{X}\mathbf{C}_{xx}^{-\frac{1}{2}}\mathbf{U}_1$ and $\hat{\mathbf{X}}_{t_1}$, the  L-CCA output after finite iterations.

**Theorem 3.** *The distance between subspaces spanned top $k_{cca}$ canonical variables of $\mathbf{X}$ and the subspace returned by L-CCA is bounded by*

$$dist(\hat{\mathbf{X}}_{t_1}, \mathbf{X}\mathbf{C}_{xx}^{-\frac{1}{2}}\mathbf{U}_1) \leq C_1 \left(\frac{d_{k_{cca}+1}}{d_{k_{cca}}}\right)^{2t_1} + C_2 \frac{d_{k_{cca}}^2}{d_{k_{cca}}^2 - d_{k_{cca}+1}^2} r^{2t_2}$$

*where $C_1$, $C_2$ are constants. $0 < r < 1$ is introduced in theorem 2. $t_1$ is the number of power iterations in L-CCA and $t_2$ is the number of gradient iterations for solving every LS problem.*

The proof of theorem 3 is in the supplementary materials.

## 5 Experiments

In this section we compare several fast algorithms for computing CCA on large datasets. First let's introduce the algorithms we compared in the experiments.

- RPCCA : Instead of running CCA directly on the high dimensional $\mathbf{X}$ $\mathbf{Y}$, RPCCA computes CCA only between the top $k_{\text{rpcca}}$ principle components (left singular vector) of $\mathbf{X}$ and $\mathbf{Y}$ where $k_{\text{rpcca}} \ll p_1, p_2$. For large $n, p_1, p_2$, we use randomized algorithm introduced in [12] for computing the top principle components of $\mathbf{X}$ and $\mathbf{Y}$ (see supplementary material for details). The tuning parameter that controls the tradeoff between computational cost and accuracy is $k_{\text{rpcca}}$. When $k_{\text{rpcca}}$ is small RPCCA is fast but fails to capture the correlation structure on the bottom principle components of $\mathbf{X}$ and $\mathbf{Y}$. When $k_{\text{rpcca}}$ grows larger the principle components captures more structure in $\mathbf{X}$ $\mathbf{Y}$ space but it takes longer to compute the top principle components. In the experiments we vary $k_{\text{rpcca}}$.

- D-CCA : See section 3.1 for detailed descriptions. The advantage of D-CCA is it's extremely fast. In the experiments we iterate 30 times ($t_1 = 30$) to make sure D-CCA achieves convergence. As mentioned earlier, when $\mathbf{C}_{xx}$ and $\mathbf{C}_{yy}$ are far from diagonal D-CCA becomes inaccurate.

- L-CCA : See Algorithm 3 for detailed description. We find that the accuracy of LING in every orthogonal iteration is crucial to finding directions with large correlation while a small $t_1$ suffices. So in the experiments we fix $t_1 = 5$ and vary $t_2$. In both experiments we fix $k_{\text{pc}} = 100$ so the top $k_{\text{pc}}$ singular vectors of $\mathbf{X}, \mathbf{Y}$ and every LING iteration can be computed relatively fast.

- G-CCA : A special case of Algorithm 3 where $k_{\text{pc}}$ is set to $0$. I.e. the LS projection in every iteration is computed directly by GD. G-CCA does not need to compute top singular vectors of $\mathbf{X}$ and $\mathbf{Y}$ as L-CCA . But by equation 1 and remark 1 GD takes more iterations to converge compared with LING . Comparing G-CCA and L-CCA in the experiments illustrates the benefit of removing the top singular vectors in LING and how this can affect the performance of the CCA algorithm. Same as L-CCA we fix the number of orthogonal iterations $t_1$ to be 5 and vary $t_2$, the number of gradient iterations for solving LS.

RPCCA , L-CCA , G-CCA are all "asymptotically correct" algorithms in the sense that if we spend infinite CPU time all three algorithms will provide the exact CCA solution while D-CCA is extremely fast but relies on the assumption that $\mathbf{X}$ $\mathbf{Y}$ both have orthogonal columns. Intuitively, given a fixed CPU time, RPCCA dose an exact search on $k_{\text{rpcca}}$ top principle components of $\mathbf{X}$ and $\mathbf{Y}$. L-CCA does an exact search on the top $k_{\text{pc}}$ principle components ($k_{\text{pc}} < k_{\text{rpcca}}$) and an crude search over the other directions. G-CCA dose a crude search over all the directions. The comparison is in fact testing which strategy is the most effective in finding large correlations over huge datasets.

**Remark 2.** *Both RPCCA and G-CCA can be regarded as special cases of L-CCA . When $t_1$ is large and $t_2$ is $0$, L-CCA becomes RPCCA and when $k_{pc}$ is $0$ L-CCA becomes G-CCA .*

In the following experiments we aims at extracting 20 most correlated directions from huge data matrices $\mathbf{X}$ and $\mathbf{Y}$. The output of the above four algorithms are two $n \times 20$ matrices $\mathbf{X}_{k_{\text{cca}}}$ and $\mathbf{Y}_{k_{\text{cca}}}$ the columns of which contains the most correlated directions. Then a CCA is performed between $\mathbf{X}_{k_{\text{cca}}}$ and $\mathbf{Y}_{k_{\text{cca}}}$ with matlab built-in CCA function. The canonical correlations between $\mathbf{X}_{k_{\text{cca}}}$ and $\mathbf{Y}_{k_{\text{cca}}}$ indicates the amount of correlations captured from the the huge $\mathbf{X}$ $\mathbf{Y}$ spaces by above four

algorithms. In all the experiments, we vary $k_{\mathrm{rpcca}}$ for RPCCA and $t_2$ for L-CCA and G-CCA to make sure these three algorithms spends almost the same CPU time ( D-CCA is alway fastest). The 20 canonical correlations between the subspaces returned by the four algorithms are plotted (larger means better).

We want to make to additional comments here based on the reviewer's feedback. First, for the two datasets considered in the experiments, classical CCA algorithms like the matlab built in function takes more than an hour while our algorithm is able to get an approximate answer in less than 10 minutes. Second, in the experiments we've been focusing on getting a good fit on the training datasets and the performance is evaluated by the magnitude of correlation captured in sample. To achieve better generalization performance a common trick is to perform regularized CCA [14] which easily fits into our frame work since it's equivalent to running iterative ridge regression instead of OLS in Algorithm 1. Since our goal is to compute a fast and accurate fit, we don't pursue the generalization performance here which is another statistical issue.

## 5.1   Penn Tree Bank Word Co-ocurrence

CCA has already been successfully applied to building a low dimensional word embedding in [6, 7]. So the first task is a CCA between words and their context. The dataset used is the full Wall Street Journal Part of Penn Tree Bank which consists of 1.17 million tokens and a vocabulary size of $43k$ [18]. The rows of $\mathbf{X}$ matrix consists the indicator vectors of the current word and the rows of $\mathbf{Y}$ consists of indicators of the word after. To avoid sample sparsity for $\mathbf{Y}$ we only consider 3000 most frequent words, i.e. we only consider the tokens followed by 3000 most frequent words which is about 1 million. So $\mathbf{X}$ is of size $1000k \times 43k$ and $\mathbf{Y}$ is of size $1000k \times 3k$ where both $\mathbf{X}$ and $\mathbf{Y}$ are very sparse. Note that every row of $\mathbf{X}$ and $\mathbf{Y}$ only has a single 1 since they are indicators of words. So in this case $\mathbf{C}_{xx}, \mathbf{C}_{yy}$ are diagonal and D-CCA can compute a very accurate CCA in less than a minute as mentioned in section 3.1. On the other hand, even though this dataset can be solved efficiently by D-CCA , it is interesting to look at the behavior of other three algorithms which do not make use of the special structure of this problem and compare them with D-CCA which can be regarded as the truth in this particular case. For RPCCA L-CCA G-CCA we try three different parameter set ups shown in table 1 and the 20 correlations are shown in figure 1. Among the three algorithms L-CCA performs best and gets pretty close to D-CCA as CPU time increases. RPCCA doesn't perform well since a lot correlation structure of word concurrence exist in low frequency words which can't be captured in the top principle components of $\mathbf{X}$ $\mathbf{Y}$. Since the most frequent word occurs $60k$ times and the least frequent words occurs only once, the spectral of $\mathbf{X}$ drops quickly which makes GD converges very slowly. So G-CCA doesn't perform well either.

Table 1: Parameter Setup for Two Real Datasets

| | PTB word co-occurrence | | | | | URL features | | | |
|---|---|---|---|---|---|---|---|---|---|
| id | $k_{\mathrm{rpcca}}$ RPCCA | $t_2$ L-CCA | $t_2$ G-CCA | CPU time | id | $k_{\mathrm{rpcca}}$ RPCCA | $t_2$ L-CCA | $t_2$ G-CCA | CPU time |
| 1 | 300 | 7 | 17 | 170 | 1 | 600 | 4 | 7 | 220 |
| 2 | 500 | 38 | 51 | 460 | 2 | 600 | 11 | 16 | 175 |
| 3 | 800 | 115 | 127 | 1180 | 3 | 600 | 13 | 17 | 130 |

## 5.2   URL Features

The second dataset is the URL Reputation dataset from UCI machine learning repository. The dataset contains 2.4 million URLs each represented by 3.2 million features. For simplicity we only use first $400k$ URLs. $38\%$ of the features are host based features like WHOIS info, IP prefix and $62\%$ are lexical based features like Hostname and Primary domain. See [19] for detailed information about this dataset. Unfortunately the features are anonymous so we pick the first $35\%$ features as our $\mathbf{X}$ and last $35\%$ features as our $\mathbf{Y}$. We remove the 64 continuous features and only use the Boolean features. We sort the features according to their frequency (each feature is a column of 0s and 1s, the column with most 1s are the most frequent feature). We run CCA on three different subsets of $\mathbf{X}$ and $\mathbf{Y}$. In the first experiment we select the $20k$ most frequent features of $\mathbf{X}$ and $\mathbf{Y}$ respectively.

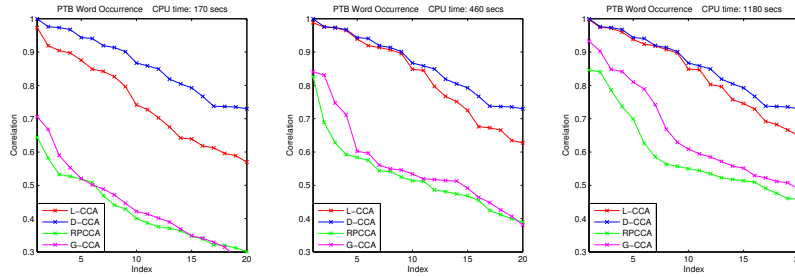

Figure 1: PTB word co-ocurrence: Canonical correlations of the 20 directions returned by four algorithms. x axis are the indices and y axis are the correlations.

In the second experiment we select $20k$ most frequent features from **X Y** after removing the top $100$ most frequent features of **X** and $200$ most frequent features of **Y**. In the third experiment we remove top $200$ most frequent features from **X** and top $400$ most frequent features of **Y**. So we are doing CCA between two $400k * 20k$ data matrices in these experiments. In this dataset the features within **X** and **Y** has huge correlations, so $\mathbf{C}_{xx}$ and $\mathbf{C}_{yy}$ aren't diagonal anymore. But we still run D-CCA since it's extremely fast. The parameter set ups for the three subsets are shown in table 1 and the 20 correlations are shown in figure 2.

For this dataset the fast D-CCA doesn't capture largest correlation since the correlation within **X** and **Y** make $\mathbf{C}_{xx}, \mathbf{C}_{yy}$ not diagonal. RPCCA has best performance in experiment 1 but not as good in 2, 3. On the other hand G-CCA has good performance in experiment 3 but performs poorly in 1, 2. The reason is as follows: In experiment 1 the data matrices are relatively dense since they includes some frequent features. So every gradient iteration in L-CCA and G-CCA is slow. Moreover, since there are some high frequency features and most features has very low frequency, the spectrum of the data matrices in experiment 1 are very steep which makes GD in every iteration of G-CCA converges very slowly. These lead to poor performance of G-CCA . In experiment 3 since the frequent features are removed data matrices becomes more sparse and has a flat spectrum which is in favor of G-CCA . L-CCA has stable and close to best performance despite those variations in the datasets.

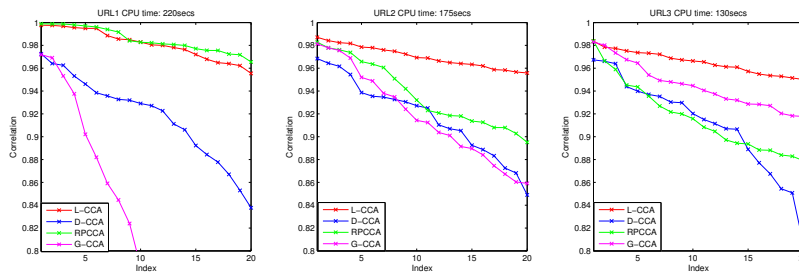

Figure 2: URL: Canonical correlations of the 20 directions returned by four algorithms. x axis are the indices and y axis are the correlations.

## 6  Conclusion and Future Work

In this paper we introduce L-CCA , a fast CCA algorithm for huge sparse data matrices. We construct theoretical bound for the approximation error of L-CCA comparing with the true CCA solution and implement experiments on two real datasets in which L-CCA has favorable performance. On the other hand, there are many interesting fast LS algorithms with provable guarantees which can be plugged into the iterative LS formulation of CCA. Moreover, in the experiments we focus on how much correlation is captured by L-CCA for simplicity. It's also interesting to use L-CCA for feature generation and evaluate it's performance on specific learning tasks.

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
