[Supplementary Material · supplementary.pdf]

# Supplementary Materials: Large scale Canonical Correlation Analysis with Least Squares

## 1. Proof of Theorem 1

Continue to use the notations from the main paper.

*Proof.* Let's focus on variable $\mathbf{X}$:
Let

$$\mathbf{A} = \tilde{\mathbf{C}}_{xy}\tilde{\mathbf{C}}_{xy}^\top = \mathbf{U}\mathbf{D}\mathbf{V}^\top\mathbf{V}\mathbf{D}\mathbf{U}^\top = \mathbf{U}\mathbf{D}^2\mathbf{U}^\top$$

and $\mathbf{B} = \mathbf{C}_{xx}^{\frac{1}{2}}\mathbf{G}$. So the columns of $\mathbf{U}$ are eigenvectors of $\mathbf{A}$. Let $\mathbf{A}^{t_1}\mathbf{B} = \mathbf{Q}_{t_1}\mathbf{R}_{t_1}$ be the QR decomposition of $\mathbf{A}^{t_1}\mathbf{B}$. Easy to check

$$\mathbf{X}_{t_1} = (\mathbf{H_X H_Y})^{t_1}\mathbf{X}\mathbf{G} = \mathbf{X}\mathbf{C}_{xx}^{-\frac{1}{2}}\mathbf{A}^{t_1}\mathbf{B} = \mathbf{X}\mathbf{C}_{xx}^{-\frac{1}{2}}\mathbf{Q}_{t_1}\mathbf{R}_{t_1}$$

Note that $\mathbf{X}\mathbf{C}_{xx}^{-\frac{1}{2}}\mathbf{Q}_{t_1}$ is an orthonormal matrix, so $(\mathbf{X}\mathbf{C}_{xx}^{-\frac{1}{2}}\mathbf{Q}_{t_1})\mathbf{R}_{t_1}$ actually gives the QR decomposition of $\mathbf{X}_{t_1}$, i.e. $\mathbf{X}_{k_{cca}} = \mathbf{X}\mathbf{C}_{xx}^{-\frac{1}{2}}\mathbf{Q}_{t_1}$.
By theorem 28.1 in (Trefethen & Bau, 1997), the columns of $\mathbf{Q}_{t_1}$ will converge to $\mathbf{U}_1$ as long as the two regularity condition hold which can be implied by our assumptions (see equation 28.4 and 28.5 in (Trefethen & Bau, 1997) for details). Therefore $\mathbf{X}_{k_{cca}} = \mathbf{X}\mathbf{C}_{xx}^{-\frac{1}{2}}\mathbf{Q}_{t_1}$ converges to $\mathbf{X}\mathbf{C}_{xx}^{-\frac{1}{2}}\mathbf{U}_1$ which are the top canonical variables of $\mathbf{X}$. The argument for $\mathbf{Y}$ is the same. $\qquad\square$

## 2. Randomized Algorithm for Finding Top Singular Vectors

Here we briefly describe a fast randomized algorithm which finds the top singular vectors of the data matrices as mentioned in section 2 of the main paper. In fact our regression based algorithms only need an orthonormal basis of the subspace spanned by the top left singular vectors of $\mathbf{X}$ and $\mathbf{Y}$ instead of the singular vectors themselves. We stick with top singular vectors in the statements and proofs of the main paper since its mathematically cleaner. All the mathematical properties of the algorithms mentioned in our paper carry through if we replace the top singular vectors with any orthonormal basis of the same subspace which is computationally more convenient because regression is a projection onto a certain subspace. Algorithm 1 is an randomized algorithm for finding this orthonormal basis, based on the idea of random subspace finder introduced by (Halko et al., 2011).

---

**Algorithm 1** Random Subspace Finder

**Input :** Matrix $\mathbf{X} \in n \times p_1$, target dimension $k_{pc}$, number of power iterations $i$.
**Output :** $\mathbf{U}_1 \in n \times k_{pc}$, an orthonormal basis of the span of the top $k_{pc}$ left singular vectors of $\mathbf{X}$, $Q_2$, an orthonormal basis of the span of the top $k_{pc}$ right singular vectors of $\mathbf{X}$
1. Generate random matrices $R_2 \in p_1 \times k_{pc}$ with i.i.d standard Gaussian entries.
2. Estimate the span of top $k_{pc}$ right singular vectors of $\mathbf{X}$ by $A_2 = (\mathbf{X}^\top\mathbf{X})^i R_2$.
3. Use QR decomposition to compute $Q_2 \in p_1 \times k_{pc}$ which is an orthonormal basis of the column space of $A_2$.
4. Compute the span of top $k_{pc}$ left singular vectors of $\mathbf{X}$ by $A_1 = \mathbf{X}Q_2$.
5. Use QR decomposition of $A_1$ to compute an orthonormal basis $\mathbf{U}_1$

---

**Remark 1.** *For numerical stability reasons, in step 2 we perform QR decomposition every time after multiply with $(\mathbf{X}^\top\mathbf{X})$ as suggested by (Halko et al., 2011). More intuitions and theoretical details of the algorithm can be found in (Halko et al., 2011).*

## 3. Gradient Descent with Optimal Step Size

Algorithm 2 gives a detailed description of the Gradient Descent algorithm we used in LING in section 3.1 of the main paper which is explained in detail by (A.Epelman, 2007).

**Remark 2.** *An implementation detail that worth mentioning is that after every iteration of GD (Algorithm 2), we actually project the coefficient $\beta_{2,t}$ onto the orthogonal complement of $V_1$ the columns of which consists top $k_{pc}$ right singular vector of $X$ (it's obtained by the randomized algorithm while computing $U_1$ as described in Algorithm 1 above), i.e. we set $\beta_{r,t+1} = \beta_{r,t+1} - V_1 V_1^\top \beta_{r,t+1}$ at the end of every iteration. The projection step significantly increases the performance of LING and our CCA algorithm. The intuition of projection step can be easily seen from the the proof of Theorem 2 (see next section) which is addressed in remark 3 of this appendix.*

**Algorithm 2** Gradient Descent with Optimal Step Size (GD)

---

**Goal :** Solve the LS problem $\min_{\beta_r \in \mathcal{R}^p} \|X\beta_r - Y_r\|^2$.
**Input :** $X \in n \times p$, $Y_r \in n \times 1$, number of gradient iterations $t_2$, an initial vector $\beta_{r,0}$ (We always initialize with 0 vector)
**Output :** $\beta_{r,t_2}$, regression coefficients after $t_2$ iterations.
**for** $t = 0$ **to** $t_2 - 1$ **do**
  $Q = 2X^\top X$
  $w_t = 2X^\top Y_r - Q\beta_{r,t}$
  $s_t = \frac{w_t^\top w_t}{w_t^\top Q w_t}$. $s_t$ is the step size which makes the target function decrease the most in direction $w_t$.
  $\beta_{r,t+1} = \beta_{r,t} + s_t \cdot w_t$.
**end for**

---

## 4. Error Analysis of LING

This section gives a detailed proof of theorem 2 in the main paper. Here we continue to use the definition and other notations in the main paper and above sections.

**Lemma 1. Sub-optimality Bound for GD**
*Assume $X$ is full rank with singular values $\lambda_1, \lambda_2..\lambda_p$. Let*

$$f(\beta_r) = \frac{1}{2}\beta_r^\top Q\beta_r - 2Y_r^\top X\beta_r + Y_r^\top Y_r$$

*be the target function value we want to minimize in regression. Assume $f^*$ is the minimum value of the target function. Let $\beta_{r,t}$ be the coefficient after $t$ iterations when initializing with $0$ vector. The sub-optimality of $\beta_{r,t}$ which is defined as $f(\beta_{r,t}) - f^*$ satisfies:*

$$\frac{f(\beta_{r,t+1}) - f^*}{f(\beta_{r,t}) - f^*} \leq \left(\frac{\lambda_{k_{pc}+1}^2 - \lambda_p^2}{\lambda_{k_{pc}+1}^2 + \lambda_p^2}\right)^2$$

*Proof.* Let $X$ have the singular value decomposition

$$X = [U_1, U_2]\begin{pmatrix} \Lambda_1 & 0 \\ 0 & \Lambda_2 \end{pmatrix}[V_1, V_2]^\top$$

where $U_1, V_1$ are top $k_{pc}$ singular vectors.

First we claim that if initialize with $0$ vector, $\beta_{r,t}, w_t$ will always be in the span of $V_2$. This is easy to see by recursion. Assume $\beta_{r,t}$ is in the span of $V_2$, by Algorithm 2, $w_t = 2X^\top Y_r - 2X^\top X\beta_{r,t}$. Note that $Y_r$ is orthogonal to $U_1$ since it's the residual of $Y$ after projecting onto $U_1$. So both $X^\top Y_r$ and $X^\top X\beta_{r,t}$ lives in the span of $V_2$ and also $w_t$ lives in the span of $V_2$. Therefore $\beta_{r,t+1} = \beta_{r,t} + s_t w_t$ also lives in the span of $V_2$. If we start with $0$ which is in the span of $V_2$, $\beta_{r,t}, w_t$ will stay in the span of $V_2$ forever.

By taking derivatives of the target function we have

$$f^* = -2Y_r^\top XQ^{-1}X^\top Y_r + Y_r^\top Y_r$$

So we have

$$
\begin{aligned}
& f(\beta_{r,t}) - f^* \\
=\ & \frac{1}{2}\beta_{r,t}^\top Q\beta_{r,t} - 2Y_r^\top X\beta_{r,t} + 2Y_r^\top XQ^{-1}X^\top Y_r \\
=\ & (Q\beta_{r,t} - 2X^\top Y_r)^\top \frac{1}{2}Q^{-1}(Q\beta_{r,t} - 2X^\top Y_r) \\
=\ & \frac{1}{2}w_t^\top Q^{-1}w_t \quad (1)
\end{aligned}
$$

$$
\begin{aligned}
& f(\beta_{r,t+1}) - f(\beta_{r,t}) \\
=\ & \frac{1}{2}(\beta_{r,t} + s_t w_t)^\top Q(\beta_{r,t} + s_t w_t) \\
& - 2Y_r^\top X(\beta_{r,t} + s_t w_t) \\
& - \frac{1}{2}\beta_{r,t}^\top Q\beta_{r,t} + 2Y_r^\top X\beta_{r,t} \\
=\ & \frac{1}{2}s_t^2 w_t^\top Q w_t \\
& + s_t w_t^\top Q\beta_{2,t} - 2s_t w_t^\top X^\top Y_r \\
=\ & \frac{1}{2}s_t^2 w_t^\top Q w_t - s_t w_t^\top w_t \\
=\ & -\frac{(w_t^\top w_t)^2}{2w_t^\top Q w_t} \quad (2)
\end{aligned}
$$

With above equation we have

$$
\begin{aligned}
& \frac{f(\beta_{r,t+1}) - f^*}{f(\beta_{r,t}) - f^*} \\
=\ & 1 - \frac{f(\beta_{r,t}) - f(\beta_{r,t+1})}{f(\beta_{r,t}) - f^*} \\
=\ & 1 - \frac{\frac{(w_t^\top w_t)^2}{2w_t^\top Q w_t}}{\frac{1}{2}w_t^\top Q^{-1}w_t} \\
=\ & 1 - \frac{(w_t^\top w_t)^2}{(w_t^\top Q w_t)(w_t^\top Q^{-1}w_t)} \quad (3)
\end{aligned}
$$

Since $w_t$ always lives in the span of $V_2$, let $z_t = V_2^\top w_t$, we have

$$\frac{(w_t^\top w_t)^2}{(w_t^\top Q w_t)(w_t^\top Q^{-1}w_t)} = \frac{(z_t^\top z_t)^2}{(z_t^\top \Lambda_2^2 z_t)(z_t^\top \Lambda_2^{-2} z_t)}$$

By Kantorovich Inequality (A.Epelman, 2007), $\frac{(z_t^\top z_t)^2}{(z_t^\top D_2^2 z_t)(z_t^\top D_2^{-2} z_t)} \geq \frac{4(\lambda_{k_{pc}+1}\lambda_p)^2}{(\lambda_{k_{pc}+1}^2 + \lambda_p^2)^2}$. Plug into equation 3 we have

$$\frac{f(\beta_{2,t+1}) - f^*}{f(\beta_{2,t}) - f^*} \leq \left(\frac{\lambda_{k_{pc}+1}^2 - \lambda_p^2}{\lambda_{k_{pc}+1}^2 + \lambda_p^2}\right)^2$$

$\square$

**Remark 3.** *From the proof it's clear that keeping the gradient $w_t$ and coefficient $\beta_{2,t}$ in the span of $V_2$ is curtail to the fast convergence to the GD algorithm. When $U_1$ consists exactly the top left singular vectors, we proved above that $w_t, \beta_{2,t}$ will always stay in the span of $V_2$. However, in practice $U_1$ in computed by Algorithm 1 which is only an approximate of the top left singular vectors. In order to compensate the error of the randomized algorithm, we project the coefficient $\beta_{2,t}$ back to the span of $V_2$ after every iteration of GD, as illustrated in remark 2 of the supplementary materials.*

### 4.1. Proof of Theorem 2

With the above lemma we can proof theorem 2 in the main paper

*Proof.* The optimality of $Y^*$ implies that $Y^* - Y$ is orthogonal to the span of $X$ and in particular is orthogonal to $\hat{Y}_{t_2} - Y^*$. By pythagorean theorem

$$\|\hat{Y}_{t_2} - Y\|^2 - \|Y^* - Y\|^2 = \|\hat{Y}_{t_2} - Y^*\|^2$$

On the other hand

$$\|\hat{Y}_{t_2} - Y\|^2 = \|(X\beta_{r,t_2} + Y_1) - (Y_r + Y_1)\|^2 = \|X\beta_{r,t_2} - Y_r\|^2$$

and

$$
\begin{aligned}
\|Y^* - Y\|^2 &= \|(Y_1 + U_2 U_2^\top Y_r) - (Y_1 + Y_r)\|^2 \\
&= \|U_2 U_2^\top Y_r - Y_r\|^2
\end{aligned}
$$

Easy to see $\|U_2 U_2^\top Y_r - Y_r\|^2 = f^*$

Put above equations and lemma 1 together,

$$
\begin{aligned}
\|\hat{Y}_{t_2} - Y^*\|^2 &= \|\hat{Y}_{t_2} - Y\|^2 - \|Y^* - Y\|^2 \\
&= \|X\beta_{r,t_2} - Y_r\|^2 - \|U_2 U_2^\top Y_r - Y_r\|^2 \\
&= f(\beta_{r,t_2}) - f^* \\
&= \left( \frac{\lambda_{k_{pc}+1}^2 - \lambda_p^2}{\lambda_{k_{pc}+1}^2 + \lambda_p^2} \right)^{2t_2} (f(\beta_{r,t_2}) - f^*)
\end{aligned}
$$

$\square$

## 5. Error Analysis of L-CCA

This section gives detailed proof of Theorem 3 in the main paper. The next lemma gives an easy way of computing distance between subspaces the proof of which is in theorem 2.6.1 (Golub & Van Loan, 1996).

**Lemma 2.** *Let*

$$\mathbf{W} = [\underset{k}{\mathbf{W}_1}, \underset{n-k}{\mathbf{W}_2}] \quad \mathbf{Z} = [\underset{k}{\mathbf{Z}_1}, \underset{n-k}{\mathbf{Z}_2}]$$

*are $n \times n$ orthonormal matrices, $dist(\mathbf{W}_1, \mathbf{Z}_1) = \|\mathbf{W}_1^\top Z_2\|_2 = \|\mathbf{W}_2^\top Z_1\|_2$*

Now let's state the key lemma for error analysis of L-CCA (below we continue to use the notation used in the main paper and supplementary):

**Lemma 3.** *Let $\mathbf{X}_t$, $\mathbf{Y}_t$ be the **LING** output in every iteration defined in Algorithm 3 of the main paper. Let $\mathbf{Y}_t = \mathbf{H_Y}\hat{\mathbf{X}}_{t-1} + \Delta_{y,t}$, $\mathbf{X}_t = \mathbf{H_X}\hat{\mathbf{Y}}_t + \Delta_{x,t}$ where $\Delta_{x,t}, \Delta_{y,t}$ denotes the error of LING compared with the exact LS solution. Assume $\|\Delta_{x,t}\|_2, \|\Delta_{y,t}\|_2 \leq \epsilon$ for every $t$. Then the distance between subspace spanned top $k_{cca}$ canonical variables and the subspace returned by L-CCA is bounded by*

$$dist(\hat{\mathbf{X}}_{t_1}, \mathbf{X}\mathbf{C}_{xx}^{-\frac{1}{2}}\mathbf{U}_1) \leq C_1 \left( \frac{d_{k_{cca}+1}}{d_{k_{cca}}} \right)^{2t_1} + C_2 \frac{d_{k_{cca}}^2}{d_{k_{cca}}^2 - d_{k_{cca}+1}^2}\epsilon$$

*where $C_1$, $C_2$ are constants.*

*Proof.* Let's focus on the $t^{\text{th}}$ iteration. Note that QR decomposition is essentially a change of basis, so we have $\hat{\mathbf{X}}_t = \mathbf{X}_t \mathbf{R}_{x,t}$ and $\hat{\mathbf{Y}}_t = \mathbf{Y}_t \mathbf{R}_{y,t}$ for some non-singular matrix $\mathbf{R}_{x,t}, \mathbf{R}_{y,t}$.

First represent $\hat{\mathbf{X}}_t$ in terms of $\hat{\mathbf{X}}_{t-1}$ and errors of LING :

$$
\begin{aligned}
\hat{\mathbf{X}}_t &= \mathbf{X}_t \mathbf{R}_{x,t} \\
&= (\mathbf{H_X}\hat{\mathbf{Y}}_t + \Delta_{x,t})\mathbf{R}_{x,t} \\
&= (\mathbf{H_X}\mathbf{Y}_t \mathbf{R}_{y,t} + \Delta_{x,t})\mathbf{R}_{x,t} \\
&= (\mathbf{H_X}(\mathbf{H_Y}\hat{\mathbf{X}}_{t-1} + \Delta_{y,t})\mathbf{R}_{y,t} + \Delta_{x,t})\mathbf{R}_{x,t} \\
&= \mathbf{H_X}\mathbf{H_Y}\hat{\mathbf{X}}_{t-1}\mathbf{R}_{y,t}\mathbf{R}_{x,t} + \mathbf{H_X}\Delta_{y,t}\mathbf{R}_{y,t}\mathbf{R}_{x,t} \\
&\quad + \Delta_{x,t}\mathbf{R}_{x,t} \quad\quad (4)
\end{aligned}
$$

Let $\mathbf{H_X}\Delta_{y,t} + \Delta_{x,t}\mathbf{R}_{y,t}^{-1} = \Delta_t$, together with equation 4 we have

$$\hat{\mathbf{X}}_t = (\mathbf{H_X}\mathbf{H_Y}\hat{\mathbf{X}}_{t-1} + \Delta_t)\mathbf{R}_{y,t}\mathbf{R}_{x,t} \quad\quad (5)$$

For simplicity assume there exist $C_0 > 1$ s.t. $\|\mathbf{R}_{y,t}^{-1}\|_2 \leq (C_0 - 1)$ for all $t$, we have

$$
\begin{aligned}
\|\Delta_t\|_2 &\leq \|\mathbf{H_X}\Delta_{y,t}\|_2 + \|\Delta_{x,t}\mathbf{R}_{y,t}^{-1}\|_2 \\
&\leq \|\Delta_{y,t}\|_2 + (C_0 - 1)\|\Delta_{x,t}\|_2 \\
&\leq C_0\epsilon \quad\quad (6)
\end{aligned}
$$

Now define $\mathbf{U}_t = (\mathbf{X}\mathbf{C}_{xx}^{-\frac{1}{2}})^\top \hat{\mathbf{X}}_t$. Since $\mathbf{X}\mathbf{C}_{xx}^{-\frac{1}{2}}$ and $\hat{\mathbf{X}}_t$ have orthonormal columns and $\hat{\mathbf{X}}_t$ lives in the column space of $\mathbf{X}$ (follows from the definition of the LING algorithm), the columns of matrix $\mathbf{U}_t$ is actually orthonormal. It's also easy to check from the definition that

$$dist(\hat{\mathbf{X}}_t, \mathbf{X}\mathbf{C}_{xx}^{-\frac{1}{2}}\mathbf{U}_1) = dist(\mathbf{U}_t, \mathbf{U}_1) \quad\quad (7)$$

From now on we can bound $dist(\mathbf{U}_t, \mathbf{U}_1)$ instead. Let $\mathbf{U} = [\mathbf{U}_1, \mathbf{U}_2]$, define

$$\mathbf{U}^\top \mathbf{U}_t = \begin{pmatrix} \mathbf{U}_1^\top \\ \mathbf{U}_2^\top \end{pmatrix} \mathbf{U}_t = \begin{pmatrix} \mathbf{W}_{1,t} \\ \mathbf{W}_{2,t} \end{pmatrix} \quad\quad (8)$$

From lemma 2, $\text{dist}(\mathbf{U}_t, \mathbf{U}_1) = \|\mathbf{W}_{2,t}\|_2$. Now let's track the quantity $\|\mathbf{W}_{2,t}(\mathbf{W}_{1,t})^{-1}\|_2$ which will eventually help us bounding $\|\mathbf{W}_{2,t}\|_2$. Recall that $\mathbf{A} = \tilde{\mathbf{C}}_{xy}\tilde{\mathbf{C}}_{xy}^\top = \mathbf{U}\mathbf{D}\mathbf{V}^\top\mathbf{V}\mathbf{D}\mathbf{U}^\top = \mathbf{U}\mathbf{D}^2\mathbf{U}^\top$.

$$\begin{pmatrix} \mathbf{W}_{1,t} \\ \mathbf{W}_{2,t} \end{pmatrix} = \mathbf{U}^\top\mathbf{U}_t$$

$$= \mathbf{U}^\top(\mathbf{X}\mathbf{C}_{xx}^{-\frac{1}{2}})^\top\hat{\mathbf{X}}_t$$

$$= \mathbf{U}^\top(\mathbf{X}\mathbf{C}_{xx}^{-\frac{1}{2}})^\top(\mathbf{H}_\mathbf{X}\mathbf{H}_\mathbf{Y}\hat{\mathbf{X}}_{t-1} + \Delta_t)\mathbf{R}_{y,t}\mathbf{R}_{x,t}$$

$$= \mathbf{U}^\top\mathbf{A}^2(\mathbf{X}\mathbf{C}_{xx}^{-\frac{1}{2}})^\top\hat{\mathbf{X}}_{t-1}\mathbf{R}_{y,t}\mathbf{R}_{x,t}$$
$$\quad + \mathbf{U}^\top(\mathbf{X}\mathbf{C}_{xx}^{-\frac{1}{2}})^\top\Delta_t\mathbf{R}_{y,t}\mathbf{R}_{x,t}$$

$$= \mathbf{U}^\top(\mathbf{A}^2\mathbf{U}_{t-1} + (\mathbf{X}\mathbf{C}_{xx}^{-\frac{1}{2}})^\top\Delta_t)\mathbf{R}_{y,t}\mathbf{R}_{x,t} \quad (9)$$

Note that

$$\mathbf{U}^\top\mathbf{A}^2\mathbf{U}_{t-1} = \mathbf{D}^2\mathbf{U}^\top\mathbf{U}_{t-1} = \begin{pmatrix} \mathbf{D}_1^2\mathbf{W}_{1,t-1} \\ \mathbf{D}_2^2\mathbf{W}_{2,t-1} \end{pmatrix} \quad (10)$$

and let

$$\begin{pmatrix} \Delta_{1,t} \\ \Delta_{2,t} \end{pmatrix} = \begin{pmatrix} \mathbf{U}_1^\top \\ \mathbf{U}_2^\top \end{pmatrix}(\mathbf{X}\mathbf{C}_{xx}^{-\frac{1}{2}})^\top\Delta_t$$

$$= \begin{pmatrix} \mathbf{U}_1^\top(\mathbf{X}\mathbf{C}_{xx}^{-\frac{1}{2}})^\top\Delta_t \\ \mathbf{U}_2^\top(\mathbf{X}\mathbf{C}_{xx}^{-\frac{1}{2}})^\top\Delta_t \end{pmatrix} \quad (11)$$

Together with equation 6 we have the following norm bound for $i = 1, 2$

$$\|\Delta_{i,t}\|_2 = \|\mathbf{U}_1^\top(\mathbf{X}\mathbf{C}_{xx}^{-\frac{1}{2}})^\top\Delta_t\|_2 \leq \|\Delta_t\|_2 \leq C_0\epsilon \quad (12)$$

because $\mathbf{U}_i$, $\mathbf{X}\mathbf{C}_{xx}^{-\frac{1}{2}}$ both have orthonormal columns. plug equation 10 11 into 9 we have

$$\begin{pmatrix} \mathbf{W}_{1,t} \\ \mathbf{W}_{2,t} \end{pmatrix} = \begin{pmatrix} \mathbf{D}_1^2\mathbf{W}_{1,t-1} + \Delta_{1,t} \\ \mathbf{D}_2^2\mathbf{W}_{2,t-1} + \Delta_{2,t} \end{pmatrix}\mathbf{R}_{y,t}\mathbf{R}_{x,t} \quad (13)$$

Equation 13 directly implies

$$\|\mathbf{W}_{2,t}(\mathbf{W}_{1,t})^{-1}\|_2$$
$$= \|(\mathbf{D}_2^2\mathbf{W}_{2,t-1} + \Delta_{2,t})(\mathbf{D}_1^2\mathbf{W}_{1,t-1} + \Delta_{1,t})^{-1}\|_2$$
$$\leq \|(\mathbf{D}_2^2\mathbf{W}_{2,t-1})(\mathbf{D}_1^2\mathbf{W}_{1,t-1})^{-1}\|_2$$
$$\quad + C_3(\|\Delta_{1,t}\|_2 + \|\Delta_{2,t}\|_2)$$
$$\leq \|\mathbf{D}_2^2\|_2\|\mathbf{W}_{2,t-1}\mathbf{W}_{1,t-1}^{-1}\|_2\|\mathbf{D}_1^{-2}\|_2 + 2C_3C_0\epsilon$$
$$= \frac{d_{k_{\text{cca}}+1}^2}{d_{k_{\text{cca}}}^2}\|\mathbf{W}_{2,t-1}\mathbf{W}_{1,t-1}^{-1}\|_2 + 2C\epsilon \quad (14)$$

where $C = C_0C_3$ are all constants independent of $t$. Note that in the first inequality, we ignore the higher order error term $\|\Delta_{1,t}\|_2 \cdot \|\Delta_{2,t}\|_2$.

Recursively apply equation 14 $t_1$ times leads to

$$\|\mathbf{W}_{2,t_1}(\mathbf{W}_{1,t_1})^{-1}\|_2$$
$$\leq \|\mathbf{W}_{2,0}(\mathbf{W}_{1,0})^{-1}\|_2\left(\frac{d_{k_{\text{cca}}+1}}{d_{k_{\text{cca}}}}\right)^{2t_1}$$
$$\quad + \sum_{j=0}^{t_1-1}\left(\frac{d_{k_{\text{cca}}+1}}{d_{k_{\text{cca}}}}\right)^{2j}2C\epsilon$$
$$= \|\mathbf{W}_{2,0}(\mathbf{W}_{1,0})^{-1}\|_2\left(\frac{d_{k_{\text{cca}}+1}}{d_{k_{\text{cca}}}}\right)^{2t_1}$$
$$\quad + \frac{d_{k_{\text{cca}}}^2}{d_{k_{\text{cca}}}^2 - d_{k_{\text{cca}}+1}^2}C\epsilon \quad (15)$$

From equation 7 we have

$$\text{dist}(\hat{\mathbf{X}}_{t_1}, \mathbf{X}\mathbf{C}_{xx}^{-\frac{1}{2}}\mathbf{U}_1) = \text{dist}(\mathbf{U}_{t_1}, \mathbf{U}_1)$$
$$= \|\mathbf{W}_{2,t_1}\|_2$$
$$\leq \|\mathbf{W}_{2,t_1}(\mathbf{W}_{1,t_1})^{-1}\|_2 \quad (16)$$

The last inequality is because $\|\mathbf{W}_{1,t_1}\|_2 \leq 1$. Put equation 15 16 together completes the proof. $\square$

Finally, use results of Theorem 2 in the main paper to bound $\epsilon$ in the above lemma directly proves Theorem 3 in the main paper.