[Reviews · NeurIPS 2014]

Submitted by Assigned_Reviewer_3

This paper introduces a fast CCA algorithm for dealing with large-scale data. Through exposing that CCA can be solved via iterative LS, a fast LS algorithm is naturally employed to speed up the CCA calculation. Error analysis for the proposed fast CCA algorithm is also provided. The experiments on large-scale datasets evaluate and compare different fast CCA methods, which are comprehensive and convincing. The paper is well written and easy to follow.

One comment on this paper is that it is not clear which theoretical results are derived in this work and which are adopted from previous works. For example, the results in Theorem 2 are similar to the results in Lu & Foster, arxiv, 2014. So it is better to explain which theoretical results are developed in this work such that the contributions of this work are clearer.

Another comment is that a critical condition for the algorithmic convergence is that the singular values of C_{xy} should be different from each other, as stated in Theorem 1. Does this condition always hold in practice? If not, how will the results be affected?
Summary: A well written and technically sound paper.

Submitted by Assigned_Reviewer_38

This paper presents an efficient solution to canonical correlation analysis (CCA). CCA is widely used in many data analysis applications, and the authors are motivated from the computational challenges of CCA, that it is slow in solving the eigenvalue problem of CCA when the matrices are huge.

The authors propose a gradient based method for addressing the challenges for computing CCA solutions. They proposed a method called LING which is a gradient based method for approximately solving CCA. The method works by performing iterative gradient steps, and QR decomposition to improve numerical stability. The proposed method is demonstrated to work well on real datasets, and the authors also provided theoretical analysis to the proposed method.

In general, this is a paper worth taking, and has solved a real practical challenge.
Summary: This paper presents a scalable solution to CCA, with theoretical and experimental evidence supporting the advantage of the method.

Submitted by Assigned_Reviewer_46

I acknowledge having read the authors' feedback. Given the clarification on the novelty aspects, I still feel that the technical contributions are minor even though the eventual algorithm is both useful and interesting. The authors claim Section 2.3 and Algortihm 1 to be novel, but several alternating algorithms for solving CCA via linear regression have been presented before (I did not mention these in the original review because I assumed Section 2 to be considered as background). Besides [20] and [22], for example Dehon, Filzmoser and Croux ("Robust methods for canonical correlation analysis", 2000), and Lykou and Whittaker ("Sparse CCA using a Lasso with positivity constraints", 2010) have in recent years presented CCA algorithms based on such derivation, citing earlier works from the 60s and 70s to motivate the re-formulation. I did not check whether all of the details are identical, but at least the relationship between the proposed algorithm and these earlier alternating LS derivations should be discussed in detail, so that readers would be able to see the novelty. If the details are the same, 2.3 cannot be considered novel.

I now understand that the LING part has novel elements as well, namely the support for sparse matrices. However, this is not at all apparent in the writing and the derivations are very similar to the UAI paper. While solving CCA is indeed an excellent use-case for that technique, it is very straightforward and LING could also be considered as background. The resulting method might well be the best choice for learning large-scale CCA models, but this would be better presented for the community by a paper that does not hide the proofs in the appendix (but instead illustrates somehow what they mean in practice) and that has much more extensive experimentation, such as explicitly showing how slow the naive algorithm is, comparison of running times on sparse and non-sparse data, better study of the conditions where RP-CCA works and fails, and illustration on how the performance of the proposed algorithm depends on the the parameters k_{pc}, t_1 and t_2 to provide practical suggestions.

(The rest is the original review)

The paper proposes an algorithm for efficient learning of CCA
for large sparse matrices, by iteratively solving learst
squares problems. Different alternatives proposed by the
authors are compared on two data sets with fairly good
results.

I am puzzled by the role of LING in this publication. It is
mentioned in the background section with a citation to a very
recent arXiv publication, yet it is explained in Section 3 as
part of the new algorithm. I would like the authors to clarify
the exact relationship between these papers. This is a
critical question since the eventual algorith is a direct
application of LING to Algorithm 1, which is not new in
itself, and the error analysis of Theorem 3 alone is not quite
sufficient to warrant publication in a top forum, especially
as it has been pushed to the Supplement.

Following the recent large-scale NLP applications of CCA, it
makes sense to consider fast alternatives. The motivation
seems to be in particular for solving setups with large p, for
which the starndard solutions are said to be quadratic or
cubic. There are, however, also solutions that are either
constant wrt p (kernel CCA solved in the dual space, for
example Hardoon et al. Neuroimage'07 uses KCCA with linear
kernels that are equivalent to CCA and p is only relevant when
computing the kernel) or linear in p (the more recent
probabilistic solutions; for example Klami et al. ICLR'14
presents a variant for sparse matrices). It would make sense
to discuss also the relationship with such algorithms, to
clarify why they would not be as fast in practice as the
proposed method (KCCA is slow for large N, whereas
probabilistic CCA should suffer from the same problem as
RP-CCA, requiring large k to capture correlations with small
variance).

I like the experiments in the sense that they attempt to
illustrate how different variants work in different
conditions. Comparison against RP-CCA is also very relevant,
since I believe for example [7] uses something like that, and
your results clearly show that RP-CCA is very good in certain
conditions (Fig. 2(a)) and poor in others. This is a valuable
result. However, I am somewhat disappointed with the way the
experiments are conducted. First of all, merely computing
correlations on the training data provides a very limited view
to CCA, especially as CCA on high-dimensional data overfits
ridiculously easily. Even though you get higher correlations
on training data compared to RP-CCA, the results could be
opposite on test data since focusing on dimensions with high
variance could reduce overlearning. This would imply that
the singular vectors are also off. Hence, I would strongly
recommend reporting also correlations computed on leave-out
test samples.

Also, the test setup where you report results for three fixed
CPU time choices does not convey a very clear message. Why
not just plot the sums of the first 20 correlations as a
function of (logarithmic) time, by running each algorithm on
multiple choices of the parameters? Then show the actual
values like in Fig 1 and 2 for one snapshot. At least for me
this would tell more about how the algoritghms work. It would
also be nice to somehow illustrate just how slow the naive
algorithms would be, perhaps by extrapolating the
computational time from a smaller data.

Quality: The method seems technically correct and the experiments
are satisfactory, albeit not optimal for illustrating the algorithms
in practice.

Clarity: The paper is easy enough to read, and the algorithm
boxes are very useful.

Originality: The method is rather straightforward and has
limited originality, since it simply plugs in LING for solving
CCA. In the end, we are merely talking about a gradient-based
algorithm for solving a linear method that is known to be
solvable with least squares algorithms.

Significance: The paper has practical signficance since there
is clear demand for CCA implementations applicable on very
large data. This would further be improved by good open-source
implementation of the algorithm, good rules of thumb for
choosing the parameters, and extensive experiments. The
theoretical significance is, however, quite limited. Maybe
some other forum would be more suitable for this kind of a
practical contribution.
Summary: Fast algorithm for computing CCA on sparse matrices of high dimensionality, with demonstrated good accuracy. The algorithm is straightforward application of LING for solving CCA, and the experiments could have been implemented better to support the lack of theoretical novely. The empirical evidence showing that RP-CCA is bad in some situations is, however, valuable, and the algorithm is likely to be of practical significance.
Author Feedback
Author rebuttal: general comments on novelty issues and connection to LING paper:

1. We didn't do a good job expositing the novel contributions of this
paper. The comments have suggested some changes of focus that will
make the contribution clearer (we will include in the final form). In particular we made a mistake by putting some novel stuff in the background section (section 2) which cause confusion about novelty issues. In other words section 2.3-2.5 and theorem 1 ,algorithm 1 are novel and shouldn't be introduced as only background knowledge. We'll definitely change that in the final form.

The main contributions of this paper: First, we formulate the
classical CCA problem as a iterative regression algorithm. This
reduction of CCA to regression is useful since there are many more
fast algorithms for regression than for CCA itself, especially in the age of "big data." This reduction contrasts with the traditional CCA algorithm which needs to whiten the data first
(this is the slow step which takes at least O(p^3) in our setup as mentioned in the introduction section), by this regression
formulation we can view CCA as Regression which is an easy to
approximate optimization problem (and we no longer need to explicitly
whitened the data). To our knowledge this is a novel idea to
approximate CCA with this LS formulation. Second, we apply a modified
version of LING to build a fast CCA solver with provable convergence
guarantee and show that our algorithm is able to catch large
correlations compare with other naive CCA approximates(all much faster than classical CCA) in the experiments. Due to space
limitation we decided to move all the proofs and some implementation
details to the appendix and focus more on providing clear pictures and
heuristics about the algorithm. We believe all the theorems in the
appendix are novel.

2. On connection to LING algorithm:

First of all, LING is only a building block of our CCA algorithm (it's
also possible to plug in other fast regression algorithms that is fits the
data well).
LING is a very recent paper published at UAI this July. By the NIPS
submission deadline it was still unpublished, that's why we spend
several paragraphs introducing LING and our hope is people can
understand the heuristic of LING from this paper alone. The LING
algorithm implemented here is a modified version to make it suitable
for sparse data matrices in our problem and we also added convergence
proof in the appendix (didn't appear in the LING paper) based on this adjusted version while the original LING paper focus more on risk calculation assuming convergence under fixed design model and didn't take into account the sparsity of the data.

assigned reviewer 3

Yes, the convergence depends on the spectrum decay and it's easier to
capture large correlations. On the other hand, when the spectrum
decays slowly, i.e. the correlations decays slowly, this means no
particular directions stands out in terms of capturing
large correlations which is a hard problem for all spectrum learning algorithms.

assigned reviewer 38

see general comments on LING and novelty issues.

assigned reviewer 46

It's true that the in sample correlation is not the best way of
evaluating CCA in machine learning sense since it may over fit. But
that suggests solving modifications to CCA problem rather than generating
a good fit to the CCA problem on training data which might or might not fit
better out of sample. In other words, with a regularizer we could avoid the over
fitting--but in a principled way. The problem would be then to
compute this regularized CCA fast on training set. Our new algorithm could be used
there also.

The problem of over fitting lead us to remove some rare words in the
penn tree bank experiment. To handle over fit you can add some
regularization to the CCA problem (equivalent to regularized every
regression in our algorithm), but that's another statistical issue.
On the other hand, our focus here is more on the mathematical side
where you are given a CCA training problem (with or without
regularization) and you want to get the best fit given a fixed CPU
time on the training dataset.

For the experiments, we tried three different groups of parameters for
each algorithm where the parameters are adjusted so that the CPU time
of all algorithms are almost the same. What you suggested is another
more condensed way of showing similar idea. Thank you for your
suggestion. We'll try to edit the experiment section a little bit if
accepted.